# LOOPED TRANSFORMERS FOR LENGTH GENERALIZATION

**Ying Fan[1], Yilun Du[2], Kannan Ramchandran[3], Kangwook Lee[1]**
[1]University of Wisconsin-Madison   [2]Massachusetts Institute of Technology   [3]UC Berkeley

## ABSTRACT

Recent work has shown that Transformers trained from scratch can successfully solve various arithmetic and algorithmic tasks, such as adding numbers and computing parity. While these Transformers generalize well on unseen inputs of the same length, they struggle with length generalization, i.e., handling inputs of unseen lengths. In this work, we demonstrate that looped Transformers with an *adaptive number of steps* significantly improve length generalization. We focus on tasks with a known iterative solution, involving multiple iterations of a RASP-L operation—a length-generalizable operation that can be expressed by a finite-sized Transformer. We train looped Transformers using our proposed learning algorithm and observe that they learn highly length-generalizable solutions for various tasks.

## 1 INTRODUCTION

Most algorithmic tasks such as coding, writing mathematical proofs, and reasoning are defined with inputs of variable *length*. The length of an input often correlates with the difficulty of the problem instance. For example, the longer the input, the more difficult the problem tends to be. We say a model perfectly *length-generalizes* if it can solve an algorithmic task on inputs of any length, even if it was only trained on data with inputs up to a finite length (Anil et al., 2022). Generally, it is hard to expect models to be trained on inputs with all possible lengths, and we need to rely on length generalization. Also, if a model can length-generalize, it means the model has truly learned the correct algorithmic solution to the task, not just a spurious solution that works only for certain lengths.

Recently, many works on Large Language Models (LLMs) have shown that we can get more powerful AI models by scaling both compute and data at training time. This scaling approach has indeed succeeded in improving accuracies on various benchmarks. However, even the largest and latest LLMs like Achiam et al. (2023) trained on much of the existing text on the Internet, still struggle with length generalization (Wu et al., 2023; Anil et al., 2022; Lee et al., 2024). One possible cause is the particular computing model. LLMs are built based mostly on the Transformer architecture (Vaswani et al., 2017). While Transformers can accept a variable length of inputs (that can be processed in parallel), they usually have a fixed depth. This might be sufficient for certain tasks, but not always.

To learn a model that can effectively generalize to longer problems, it is important to consider architectures that can adaptively adjust the computational budget to the difficulty of the tasks (Anil et al., 2022; Du et al., 2022; 2024). One approach to achieve this is to explicitly generate intermediate output tokens, similar to writing down a scratchpad, which improves LLMs' capability for solving harder problems (Nye et al., 2021). In theory, LLMs may generate more scratchpad tokens representing intermediate computation when solving a more difficult task, indicating that they can allocate elastic computation according to the length and difficulty of the given instance. This approach can be learned by explicitly training a model on data with intermediate computation steps (Ling et al., 2017; Cobbe et al., 2021). Alternatively, it can be achieved via Chain-of-Thought (CoT) reasoning with few-shot examples (Wei et al., 2022) or even in a zero-shot manner (Kojima et al., 2022). Notice that these approaches still use fixed-depth models. While these approaches help solve more complex reasoning tasks, they are still far from achieving near-perfect length generalization for simple algorithmic tasks. For instance, Lee et al. applied CoT for arithmetic tasks but observed that Transformers cannot length generalize even for simple addition tasks (Lee et al., 2024).

Recently, there has been growing interest in using recurrent architectures for reasoning (Dehghani et al., 2018; Bai et al., 2019; Bansal et al., 2022; Yang et al., 2024). Unlike standard RNN-type

architectures that process the input sequence incrementally, one can consider a recurrent architecture that processes the entire input sequence multiple times, passing the intermediate output to the next iteration's input, possibly along with the original input. In particular, if the base model in each iteration is a Transformer, this model is called a Looped Transformer (Yang et al., 2024).

Looped Transformer can naturally break the limitation of the fixed depth in the standard Transformer architecture: *One can adjust the number of looped steps based on the computational complexity of the underlying algorithmic solution*. Consider a problem set where 1) The problems can be solved by a loop of one RASP-L (Zhou et al., 2024a) program[1], i.e., each step in the loop can be performed by a decoder-only Transformer with a fixed depth; 2) The number of steps needed in the loop depends on the problem's complexity, i.e., more difficult problems could potentially require more steps to solve. Under the length generalization scheme, we consider the number of steps depending on the problem length, and define this problem set as $n$-RASP-L problems. For $n$-RASP-L problems, if we can learn these length-independent steps, we can utilize an adaptive number of steps to achieve length generalization.

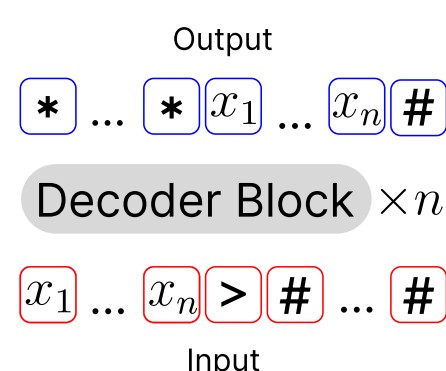

Figure 1: **Method Overview.** During training, we supervise the output of the model to match the target data only after a certain number of steps of applying the same decoder block, helping the model learn intermediate steps that can be reused and can handle input of arbitrary lengths. All grey blocks share the same parameters. Examples are from the Copy task with $n$ symbols. "#" indicates EOS, "*" indicates ignored output, and ">" indicates the end of the query (EOQ).

Inspired by this observation, we study training Looped Transformers models for length generalization. Specifically, we consider a training setup where we do not require any intermediate supervision data (such as reasoning steps or scratchpad). We only assume access to end-to-end supervision (input and output) and the number of steps needed. Depending on the number of steps, we iteratively apply the same decoder block and then decode the final answer; See Figure 1 for illustration. At inference time, the model could either decide when to stop with predefined stopping criteria or stop when reaching the ground-truth number of steps. Empirically, we show that looped Transformers with an adaptive number of steps can successfully length-generalize to longer lengths simply by appropriately adapting the number of loops at inference time, indicating that our approach encourages the model to implicitly learn the necessary steps to solve a task.

Our contributions can be summarized as follows: **(1)** We first formally define $n$-RASP-L problems, and provide examples of $n$-RASP-L solutions to the Copy, Parity, and Addition tasks (Section 3); **(2)** We propose to learn $n$-RASP-L problems with Looped Transformers where we supervise the final answer in a step-dependent way, which enables us to use an adaptive number of steps depending on the problem complexity (Section 4); **(3)** Empirically, we show that our proposed method outperforms the baseline approaches in terms of length generalization performance (Section 6).

## 2 BACKGROUND

### 2.1 RASP-L

A decoder-only Transformer is a type of Transformer architecture that consists of only the decoder part of the original Transformer model introduced by Vaswani et al. (2017), where a causal mask is applied to the attention weights to prevent the model from attending to future tokens.

RASP (Restricted Access Sequence Processing) (Weiss et al., 2021) is a computational model for the Transformer architecture in the form of a programming language. RASP-L (Zhou et al., 2024a), is a learnable subset of the RASP language. Some key points about RASP-L are:

- RASP-L programs accept an input sequence and return an output sequence of the same length for an *arbitrary length*, like decoder-only Transformers.

---

[1]Here we consider a more general way to loop, i.e., predicting all missing tokens at the end of the loop, not necessarily in the way of predicting the single next token at a time. See more discussions in Section 2.2.

- The core operations in RASP-L include element-wise operations on sequences and a specific type of non-elementwise operation called `kqv`, which simulates a causal attention layer.
- RASP-L has restrictions on the allowed operations to ensure learnability: It does not allow arbitrary index arithmetic, and restricts operations on token indices to order comparisons and computing successor/predecessor.
- RASP-L *does not allow control flow* statements like branching or loops. Programs must be straight-line code, with each line being a call to a core function or another RASP-L program.

In Zhou et al. (2024a), they show algorithmic tasks that can be written as a RASP-L program can be easily learned by a Transformer in a length-generalizable way with next-token prediction. The length-generalizable tasks include counting, finding the mode, copying the input sequence (consisting of unique tokens), and sorting. However, they also showed that for algorithmic tasks whose RASP-L program representation is not known to exist, such as addition, parity, and copying the input sequence, it is hard to learn in a length-generalizable way. In other words, once the Transformer is trained on in-distribution data up to a particular length, it fails to generalize to unseen lengths.

## 2.2 NEXT-TOKEN PREDICTION AND FULL-OUTPUT PREDICTION

Decoder-only Transformers are naturally convenient for next-token prediction (NTP) which could be efficiently trained in parallel. In Zhou et al. (2024a), their setup and RASP-L solutions are both constrained to predicting the single next token: During training, the full sequence (both the query and the answer) is provided as input and the output is expected to be the shifted sequence. During inference, only the query part is provided, and the model continues to output the next token and append the token to the current sequence until the output token is EOS. The output locations before the end of query (EOQ) sign are ignored. See (a) in Figure 2 for illustration.

On the other hand, we can also consider a more general way of predicting the answer: full-output prediction (FOP). During both training

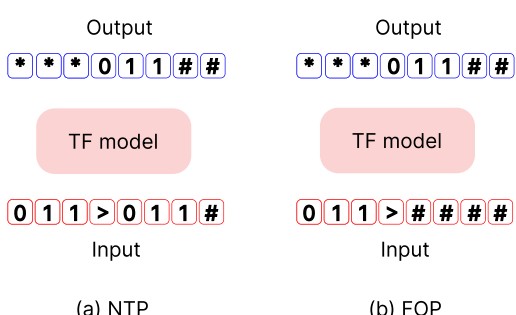

Figure 2: Visualization of the next-token prediction (NTP) and full-output prediction (FOP) schemes. "#" indicates EOS, "*" indicates ignored output, and ">" indicates the end of the query (EOQ).

and inference time, the input given is just the query part, and the rest of the locations are filled with multiple EOS tokens to keep the input and the output to be the same length. The model is supposed to output the answer with a shifted location, and the output locations before the EOQ sign are ignored; see (b) in Figure 2. Notice that in FOP, the model is not forced to predict token-by-token as NTP. Instead, the model is expected to predict all missing tokens after all internal processing steps.

# 3 $n$-RASP-L

Recall that RASP-L programs do not allow loops. If we consider the next-token prediction (NTP) scheme, it means that we need to find the same RASP-L program (which can be represented with a fixed-depth decoder-only Transformer) to predict the next token given any possible prefix in the answer sequence. Such solutions might not always exist for all problems: there is no known RASP-L program for addition, parity, and copy under the NTP scheme (Zhou et al., 2024a).

On the other hand, architectures such as the Looped Transformer have external loops embedded in the architecture which naturally provides adaptive depth. Thus, a natural question is: what kind of algorithmic tasks can we represent with a decoder-only Transformer in a loop? Specifically, what if we also allow the number of iterations to explicitly depend on the input length, say $n$? Moreover, what if we are not constrained by the NTP scheme, but a more general FOP scheme?

Inspired by these questions, we define the following class of algorithmic tasks:

**Definition 3.1** ($n$-RASP-L). *A program $P$ is called an $n$-RASP-L program if (1) there exist $T : \mathbb{N} \to \mathbb{N}$, and (2) $P$ can be decomposed to a sequential application of $P'$ for $T(n)$ steps with a possible pre-processing step $P_{pre}$ and post-processing step $P_{post}$: $P = P_{pre} \circ (P')^{T(n)} \circ P_{post}$ where $P', P_{pre}, P_{post} \in RASP\text{-}L$.*

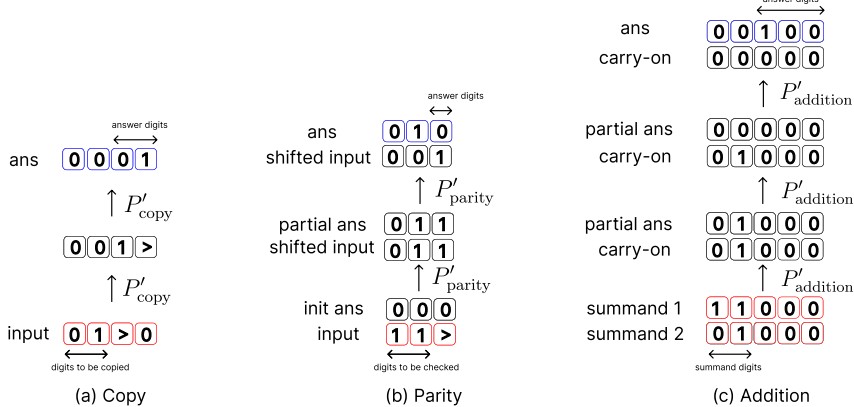

Figure 3: Visualization of the $n$-RASP-L solutions for Copy, Parity, and Addition with $n = 2$. Copy is implemented by $n$ iterations of shifting; Parity is implemented by $n$ iterations of shifting and XOR; Addition is implemented by $n + 1$ iterations of shifted XOR and AND; The inputs are preprocessed. See details in Section 3.

We show that $n$-digit addition, $n$-bit parity, copying $n$ symbols indeed have $n$-RASP-L solutions.

**Proposition 3.2.** *(Parity.) There exists a n-RASP-L program with $T(n) = n$ that solves the $n$-bit parity check task:*

$$\underbrace{\boxed{x_1}\,\boxed{\dots}\,\boxed{x_n}\,\boxed{>}}_{n \text{ tokens}}\ \underbrace{\boxed{\#}\,\boxed{\dots}\,\boxed{\#}}_{n' \text{ tokens}, n' \geq 0} \Rightarrow \underbrace{\boxed{*}\,\boxed{\dots}\,\boxed{*}}_{n \text{ tokens}}\,\boxed{y}\,\underbrace{\boxed{\#}\,\boxed{\dots}\,\boxed{\#}}_{n' \text{ tokens}},$$

*where $y$ is the parity check result for the arbitrary binary input sequence $\{x_i\}$.*

*Proof.* See Listing 1 in Appendix A, where the number of steps required in `parity_loop` is $T(n) = n$ for the input query with $n$ bits. $\square$

**Proposition 3.3.** *(Copy.) There exists a n-RASP-L program with $T(n) = n$ that solves the $n$-symbol copy task:*

$$\underbrace{\boxed{x_1}\,\boxed{\dots}\,\boxed{x_n}\,\boxed{>}}_{n \text{ tokens}}\ \underbrace{\boxed{\#}\,\boxed{\dots}\,\boxed{\#}}_{n' \text{ tokens}, n' \geq n-1} \Rightarrow \underbrace{\boxed{*}\,\boxed{\dots}\,\boxed{*}}_{n \text{ tokens}}\,\underbrace{\boxed{x_1}\,\boxed{\dots}\,\boxed{x_n}}_{n \text{ tokens}}\,\underbrace{\boxed{\#}\,\boxed{\dots}\,\boxed{\#}}_{n' - n + 1 \text{ tokens}},$$

*where $\{x_i\}$ denotes arbitrary binary input symbols.*

*Proof.* See Listing 2 in Appendix A, where the number of steps required in `copy_loop` is $T(n) = n$ for the input query with $n$ symbols. $\square$

**Proposition 3.4.** *(Addition.) There exists a n-RASP-L program with $T(n) = n + 1$ that solves the $n$-digit addition task:*

$$\underbrace{\boxed{x_1}\,\boxed{\dots}\,\boxed{x_n}}_{n \text{ tokens}}\,\boxed{+}\,\underbrace{\boxed{y_1}\,\boxed{\dots}\,\boxed{y_n}}_{n \text{ tokens}}\,\boxed{>}\ \underbrace{\boxed{\#}\,\boxed{\dots}\,\boxed{\#}}_{n' \text{ tokens}, n' \geq n} \Rightarrow \underbrace{\boxed{*}\,\boxed{\dots}\,\boxed{*}}_{2n + 1 \text{ tokens}}\,\underbrace{\boxed{z_1}\,\boxed{\dots}\,\boxed{z_{n+1}}}_{n + 1 \text{ tokens}}\,\underbrace{\boxed{\#}\,\boxed{\dots}\,\boxed{\#}}_{n' - n \text{ tokens}},$$

*where $\{x_i\}$, $\{y_i\}$ are arbitrary binary summands and $\{z_i\}$ denotes the result of adding $\{x_i\}$ and $\{y_i\}$[2].*

*Proof.* See Listing 3 in Appendix A, where the number of steps required in `addition_loop` is $T(n) = n + 1$ for the input summands with $n$ digits each. $\square$

We present visualizations of the intermediate steps in the loops of our n-RASP-L solutions in Figure 3: For the parity task, $P'_{\text{parity}}$ is to shift the input sequence to the right by 1 and calculate XOR of the answer sequence and the input sequence; For the copy task, $P'_{\text{copy}}$ is to shift the input sequence to the right by 1; For the addition task $P'_{\text{addition}}$ is to calculate the XOR of two sequences and shift the results to the right by 1 position as the partial answer, and calculate the AND of two sequences as the carry-on sequence[3].

---

[2]For simplicity, we include the leading 0's to keep the same length of the output for all possible inputs.

[3]Here we omit the pre-processing and post-processing steps like handling EOS ("#") and EOQ (">") tokens which can be done by fixed-depth attention layers outside of the loop (see Listings 1, 2, 3).

## 4 LEARNING $n$-RASP-L PROBLEMS WITH LOOPED TRANSFORMERS

Consider a task solvable by an $n$-RASP-L program. It is straightforward to learn the looped Transformer model with the supervision of the ground truth intermediate outputs: One can use a fixed-depth TF block and simply supervise the input and output for each step. However, such intermediate supervision can be difficult to get, just like collecting helping CoT steps could be difficult. Therefore, a more interesting setup we consider is to learn looped Transformers in an end-to-end manner *without* intermediate-step supervision.

Here we present a novel framework for length generalization: In the absence of ground truth CoT data/intermediate output, we propose to leverage the inherent structure of the problem with the help of "knowing when to stop". We present the setup for training data in Section 4.1, the model architecture and training algorithm in Section 4.2, and the inference algorithm in Section 4.3.

### 4.1 END-TO-END SUPERVISED DATA WITHOUT INTERMEDIATE STEP SUPERVISION

We consider the following settings for the training data and the tasks:

- There exists an $n$-RASP-L program that solves the given task.
- Training data consists only of $(x, y)$ pairs, but not intermediate steps. That is, we do not have access to $P'(x), P'(P'(x)), \ldots$.
- $T(n)$, i.e., the pre-defined number of iterations to solve the problem (with some $P'$) is available in the training data[4].
- The length $n$ is diversely distributed in the dataset, e.g., $n \in \{1, \ldots, n_{\max}\}$ where $n_{\max}$ is the maximum number of lengths in the dataset; The pre-defined number of steps needed $T(n)$ is also diversely distributed in the dataset, e.g., $T(n) \in \{T(1), \ldots, T(n_{\max})\}$ where $T(n_{\max})$ is the maximum number of steps in the dataset[5].

### 4.2 LOOPED TRAINING WITH STEP SUPERVISION

#### 4.2.1 ARCHITECTURE OF THE LOOPED TRANSFORMERS

We present the architecture for Looped Transformer model in Figure 1. The key characteristics are:

**Recurrence:** Instead of having a simple stack of blocks, the Looped Transformer is recurrent (like Giannou et al. (2023) but with decoder-only structure) in the sense that we reuse the same decoder block (which consists of a certain number of layers) for a number of looped steps, and we can adjust the number of looped steps at will.

**Input injection:** For each step, the original input sequence is injected together with the output from the previous decoder block, i.e. the input embeddings are added to the output embeddings of the previous step as the input of the current step. With input injection, the model can maintain a strong connection to the original input, preventing information loss with improved performance (Bai et al., 2019; Yang et al., 2024).

**Positional embeddings:** Notice that there is no positional encoding in the RASP-L operations (Zhou et al., 2024a). To follow our $n$-RASP-L assumption and test the effect of the looped training only, we use NoPE (Kazemnejad et al., 2024) in decoder-only Transformers to avoid the impact from different positional embeddings[6].

#### 4.2.2 TRAINING ALGORITHM

Given a dataset $D = \{(\{(x_l)_{l=1}^{L_i}\}_i, \{(y_l)_{l=1}^{L_i}\}_i, T_i, L_i)\}_{i=1}^N$, where $\{(x_l)_{l=1}^{L_i}\}_i$ is the input with $L_i$ tokens, $\{(y_l)_{l=1}^{L_i}\}_i$ is the output with $L_i$ tokens, and $T_i$ is pre-defined number of steps of sample $i$. We aim to learn the transformer model $M_\theta$[7] by minimizing the following loss:

---

[4]This assumption is to provide supervision for when to stop during training; for inference, we can either use the pre-defined steps or leverage the confidence of the output as a stopping criterion (see Section 4.3 for details.)

[5]The length of the problem is not necessarily the same as the actual length of the input due to EOS and EOQ tokens; see Section 6.1.1 for the definition of the length of the specific tasks.

[6]NoPE is shown to inherently learn to use relative positional embeddings in practice Kazemnejad et al. (2024).

[7]$M_\theta$ only handles the embedding space and we use greedy decoding to get the decoded output.

$$\mathbb{E}_D[\mathcal{L}\left(f_{T_i}(M_\theta, \{(\{(x_l)_{l=1}^{L_i}\}_i), \{(y_l)_{l=1}^{L_i}\}_i\right)], \tag{1}$$

where $\mathcal{L}$ is the cross entropy loss and $f_{T_i}(M_\theta, \{(x_l)_{l=1}^{L_i}\}_i) = \underbrace{M_\theta(M_\theta(\cdots M_\theta(\{(x_l)_{l=1}^{L_i}\}_i)))}_{T_i \text{ iterations}}$.

How would our training help to learn more length-generalizable steps without intermediate supervision? Consider that for some input, we want to match the target after $T$ steps, $T \geq 1$. Although we do not have intermediate supervision for this input, there could exist some other input where we match the output after $T - 1$ steps as shown in Figure 1, which means the model also receives supervision from $T - 1$ steps with a different input length. The same argument applies to other $T$ values covered in the training set, so the same decoder block receives supervision from not only variable lengths, but also a variable number of steps. As a result, we expect the model to learn length-generalizable intermediate steps that could potentially generalize to more steps at test time.

### 4.3 Adaptive inference

Recall that looped Transformers can use adaptive depth at inference time, so we need certain rules to decide when to stop. In this paper, we consider two rules: 1) *Oracle*: We can assume that the number of steps needed is given; 2) *Maximum confidence*: We can use confidence base rules to decide when to stop, i.e., stop when we are confident about the output at the current step. More specifically, for 2), given $B$ test sequences with length $L$: $\{(x_l)_{l=1}^{L}\}_{i=1}^{B}$ and a trained model $M_\theta$, we can get the number of steps $T$ from Equation (2):

$$T = \arg\min_{t\in[1,T_{\max}]}\mathcal{L}\left(f_t(M_\theta, \{(x_l)_{l=1}^{L}\}_{i=1}^{B}), \{(\hat{y}_l^t)_{l=1}^{L}\}_{i=1}^{B}\right), \tag{2}$$

where $\{(\hat{y}_l^t)_{l=1}^{L}\}_{i=1}^{B}$ is the decoded sequences from $f_t(M_\theta, \{(x_l)_{l=1}^{L}\}_{i=1}^{B})$ at step $t$, $T_{\max}$ is the maximum number of steps. Here $B$ could be $N_{\text{test}}$, which is the size of the test set with some specific length. Also, we can choose $B = 1$ to calculate the per-sample stopping criterion (see Section 6.4).

## 5 Related work

**Positional embedding for length generalization.** Positional embeddings have been shown to greatly affect Transformers' ability to generalize to longer lengths (Press et al., 2021; Kazemnejad et al., 2024; Ruoss et al., 2023; Su et al., 2024; Li et al., 2023; Cho et al., 2024; Sabbaghi et al., 2024; McLeish et al., 2024; Golovneva et al., 2024). By designing positional embedding schemes that better capture relative positional information with techniques such as randomization and functional representations, researchers have made significant progress in improving length generalization. Especially, Cho et al. (2024) and McLeish et al. (2024) use tailor-made positional embeddings for some arithmetic problems without potential generality. [8] This direction is orthogonal to our work since there is no positional encoding in RASP-L operations. We choose no positional embedding in our experiments, but other positional embeddings could further be synergistically applied with our approach. However, they might not be expressed as RASP-L operations. We leave further investigation with different positional embeddings to future work.

**RNNs and Chomsky Hierarchy.** Delétang et al. (2022) conduct an extensive empirical study to investigate the limits of the generalization performance of different neural network structures, demonstrate that grouping tasks according to the Chomsky hierarchy allows forecasting whether certain architectures will be able to generalize to out-of-distribution inputs. Their results show that RNNs and Transformers fail to generalize on non-regular tasks, LSTMs can solve regular and counter-language tasks, and only networks augmented with structured memory (such as a stack or memory tape) can successfully generalize on some context-free and context-sensitive tasks. In our paper, the Looped Transformer architecture also has augmented memory and the recurrent structure but is potentially more powerful since each iteration contains an operation of the whole sequence.

---

[8] In McLeish et al. (2024), they show that models with weight-tied layers (but with a fixed depth) can improve the generalization ability when comparing with the variants of the same positional embedding, but they do not find adaptive depths to be helpful since they do not perform the step-specific training as our method, while the key to our method is to use models with adaptive depths. To also compare with this baseline, we add NTP-Loop in Section 6.1.2.

| Method | Encoder/Decoder | Prediction Type | PE | Input Injection | Halting Mechanism |
|--------|-----------------|-----------------|-----|-----------------|-------------------|
| UT | Both | NTP | Yes | No | ACT (Bolukbasi et al., 2017) |
| PonderNet | Both | NTP | Yes | No | Halting node |
| Ours | Decoder-only | FOP | No | Yes | Confidence based or predefined |

Table 1: Comparison between UT, PonderNet, and ours. PE is short for "Positional Embeddings".

**Universal Transformers and other looped models.** Our method is highly inspired by Universal Transformers (UT) (Dehghani et al., 2018), but we introduce several novel modifications to design looped Transformers that are compatible with our $n$-RASP-L assumption. One major architectural innovation is the use of FOP, while all the other prior works are based on NTP. We also only use decoder-only Transformers, which is different from UT and the follow-up work PonderNet (Banino et al., 2021), which use both encoder and decoder Transformers. In addition to these two critical differences, we do not use any positional encoding, and use a simpler halting mechanism. Moreover, we find input injection useful to further improve the performance (see details in Section 6.3). Table 1 summarizes the differences between ours and the previous approaches. Besides architectural differences, we are also the first to show the benefit of using step-dependent supervision for training looped Transformers. Apart from Transformers, Bansal et al. (2022) study learning recurrent networks to generalize to harder maze problems than seen during training, but with a focus on CNNs.

**Input representation.** Recall that adding two numbers of length $n$ could not be solved by a RASP-L program where the difficulty mainly comes from indexing operations (Zhou et al., 2024a). It could be solved by reformatting the input so that each digit is presented to the model with "index hints" in Zhou et al. (2024a). Such reformatting enables a simple RASP-L program for addition. Similarly, representing the answer in reversed order also helps because the corresponding RASP-L program gets much shorter, providing a concrete justification of the empirical observation made in Lee et al. (2024). However, such input representations are highly dependent on the specific problems and might not necessarily exist in general.

**COT.** Scratchpad or CoT reasoning (Nye et al., 2021; Ling et al., 2017; Cobbe et al., 2021; Wei et al., 2022; Hou et al., 2024) is also useful for length generalization as it could simplify the next-token prediction task with intermediate results presented to the input layer. There are also potential drawbacks and limitations to CoT reasoning. First, CoT training data could be hard to collect. Training and inference with pause tokens (Goyal et al., 2023) has been proposed to learn implicit CoT steps without CoT data, but pause tokens only increase horizontal compute, not sequential compute. Second, not all CoT steps are helpful. If CoT steps introduce additional complexity or require operations not easily expressible in RASP-L, then CoT may hinder length generalization, as shown in Zhou et al. (2024a). Moreover, CoT steps that could convert the next token prediction task to RASP-L programs might not always exist. Besides, CoT is normally constrained to fixed-depth models, while we study a more general and powerful way to use adaptive compute at inference time.

## 6 Experiments

We evaluate the efficacy of looped Transformers in solving tasks that require length generalization. We introduce the experimental setup in Section 6.1, present length generalization results in Section 6.2 and ablation studies in Section 6.3, and visualize the stopping criterion in Section 6.4. Code is available at `https://github.com/UW-Madison-Lee-Lab/looped-tf`.

### 6.1 Experimental setup

#### 6.1.1 Tasks

Here we consider tasks with $n$-RASP-L solutions presented in Section 3: Parity, Copy, and Addition, together with more tasks like calculating the sum, multiplication, and calculating the unique set.

**Parity.** Checking the parity of the binary string. Example input: `0` `0` `0` `1` `1` `>` `#` `#`, example output: `*` `*` `*` `*` `*` `0` `#` `#`. We define the length of the problem to be the number of the digits, set $T$ (the number of steps needed) to be the same as the length, and train with length $[1, 20]$.

**Copy (with repeated tokens).** Copying the binary string. Example input: `1` `0` `1` `>` `#` `#` `#` `#`, example output: `*` `*` `*` `1` `0` `1` `#` `#`. We define the length of the problem to be the number of

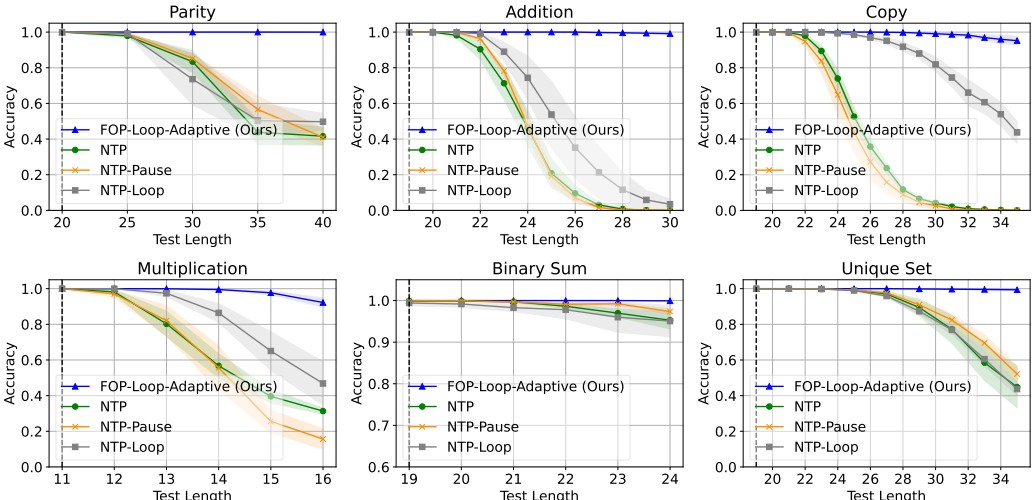

Figure 4: **Length Generalization Performance.** Our looped Transformer model with adaptive depth generalized better than NTP methods across studied tasks, including the variants with pause tokens and weight-tied layers. The vertical dashed line indicates the maximum training length.

the binary digits to copy, set $T$ to be the same as the problem length, and train with length $[1, 20]$. It has been shown that copy with unique tokens could be easily solved by inductive head (Olsson et al., 2022), but copy with repeated tokens (e.g., binary) does not length-generalize with vanilla NTP training (Zhou et al., 2024a).

**Binary Addition.** Performing binary addition of two binary numbers with the same number of digits, and the output has one more digit (without removing leading 0 if it appears). Example input: `1` `0` `+` `1` `1` `>` `#` `#` `#`, example output: `*` `*` `*` `*` `*` `1` `0` `1` `#` `#`. We highlight that we do not reverse the output like recent works (Lee et al., 2024; McLeish et al., 2024; Zhou et al., 2024b). We define the length of the problem to be the number of digits to be added, set $T$ to be the same as the problem length, and train with length $[1, 20]$. It has been shown that binary addition without index hint is hard to generalize in vanilla NTP (Zhou et al., 2024a).

**Binary Sum.** Calculating the sum of the binary string in the binary form (in reversed order). Example input: `1` `0` `1` `1` `>` `#` `#` `#` `#`, example output: `*` `*` `*` `*` `1` `1` `#` `#` `#`. We define the length of the problem to be the number of binary digits to be added, set $T$ to be the same as the problem length, and train with length $[1, 20]$.

**Binary Multiplication.** Multiplying two binary numbers, while the first number has up to two digits. The output is in reversed order and the length is the sum of the lengths of two numbers, without removing leading 0. Example input: `1` `1` `×` `1` `1` `0` `>` `#` `#` `#` `#` `#`, example output: `*` `*` `*` `*` `*` `*` `0` `1` `0` `0` `1` `0` `#`. We define the problem length to be the length of the second number, and set $T$ to be the product of the lengths of two numbers, and train with length $[1, 12]$.

**Unique Set.** Calculating the unique set with the first occurrence order with an alphabet of 50 tokens. Example input: `1` `4` `2` `2` `4` `3` `>` `#` `#` `#` `#` `#`, example output: `*` `*` `*` `*` `*` `*` `1` `4` `2` `3` `#` `#`. We define the length of the problem to be the number of digits to be calculated, set $T$ to be the same as problem length, and train with length $[1, 20]$.

### 6.1.2 BASELINE METHODS

**Vanilla NTP.** We use vanilla next-token prediction as a baseline, referred to as "NTP" in Figure 4. To ensure that the baseline method uses a maximum effective depth comparable to ours during training, we train the transformer model with a depth 20 times the depth of the looped block in our approach.

**NTP with pause tokens.** Training and inference with pause tokens (Goyal et al., 2023) is a way to implicitly learn implicit CoT steps without CoT data by enabling extra compute pathways before outputting the answer in NTP. We use it as a baseline with the same depth as in vanilla NTP, referred to as "NTP-Pause" in Figure 4. We include a visual illustration of NTP-Pause in Figure 8 in Appendix D.

**NTP with weight-tied layers.** Using weight-tied layers but with a fixed number of overall depths in NTP is also shown to improve the performance in McLeish et al. (2024). Here we fix the number of looped steps as 20, use the same depth as the decoder block of our looped model, and train the model with NTP as another baseline which is referred to as "NTP-Loop" in Figure 4.

### 6.1.3    TRAINING AND EVALUATION SETUP

For training, we use a decoder-only GPT-2 architecture (Radford et al., 2019). We adopt a curriculum learning strategy for all methods that starts from the smallest length and incrementally increases the length during training till it reaches the maximum length as in Garg et al. (2022).

For evaluation, we measure the exact match accuracy for the whole output sequence. For our looped inference, we test two possible stopping criteria discussed in Section 4.3: 1) *Oracle*: Adopt the same rule when generating the dataset as the number of steps to perform, 2) *Maximum confidence*: Run a maximum number of steps, and choose the step using Equation (2)[9]. We report test results from 1) in Section 6.2 and 6.3, and we also find 2) to be an effective stopping criterion in Section 6.4.

Full details of training and evaluation are in Appendix F.

### 6.2    LENGTH GENERALIZATION RESULTS

We present the generalization performance on various reasoning tasks in Figure 4.

**Looped Transformers help with length generalization.** Our looped training significantly improves the length generalization performance. For example, for Parity, it can generalize to more than 40 digits near perfectly when trained with up to 20 digits. Moreover, for tasks like addition and copy, where the next token prediction failed when tested on maximum training length $+10$, our looped model can still perform almost perfectly.

**Variants of NTP could improve generalization but not as effectively as our adaptive-depth model.** Compared with vanilla NTP, we observe that NTP-Loop could lead to improved generalization in tasks like Addition, Copy and Multiplication. Similarly, NTP-pause could introduce slight improvement in Binary Sum and Unique Set. However, they all fall behind compared with our method.

### 6.3    ABLATION STUDIES

In Section 6.2, we compare with NTP baselines while the efficacy of components in our architecture design remains unclear. In this section, we compare with FOP variants of our model in Figure 7 (Appendix C): "FOP-Loop-Adaptive-WO" indicates our method but without input injection; "FOP-Pause" indicates FOP with pause tokens[10]; "FOP" indicates vanilla FOP without weight-tied layers and adaptive depths.

**Effect of input injection.** We observe the generalization performance with input injection is generally better than without it, which aligns with the findings in Bai et al. (2019) and Yang et al. (2024). The effect of input injection is more visible in tasks like Addition, Binary Sum, and Unique Set.

**Comparison with pause tokens and vanilla FOP.** Training with pause tokens in FOP could sometimes boost the generalization performance compared to vanilla FOP, but not as effective as our method with looped steps and adaptive depth. As discussed in Goyal et al. (2023), pause tokens mainly introduce parallel but not sequential compute, which is less powerful than adaptive depth. Besides, we find worse in-distribution accuracy for both FOP and FOP-Pause in Addition, which mainly comes from the difficulty in training a deep model ($20\times$ the depths of the decoder block used in the looped model) in FOP. It further highlights the importance of supervision with variant depths used in our training.

### 6.4    THE STOPPING CRITERION AND VISUALIZATIONS

In this section, we visualize the accuracy and the cross-entropy loss with respect to the decoded output in each iterative step across tasks in Figure 5. We also provide more visualizations from other test lengths in Appendix B. The vertical lines in Figure 5 are chosen based on the full test set using

---

[9]Another option is to set a threshold for the cross-entropy loss and stop when the threshold is first met. This will also succeed if the maximum confidence rule works.

[10]Visual illustration of FOP-Pause is in Figure 9 in Appendix D.

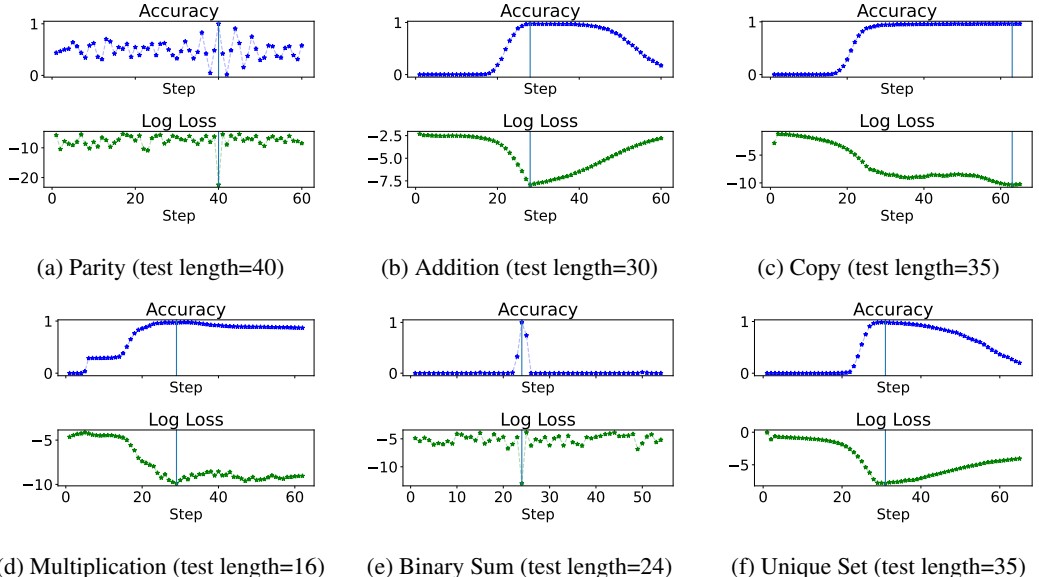

Figure 5: **Stopping criterion visualizations.** Plot of the stopping criterion on the test set. The vertical line indicates the step chosen from Equation (2) (where $B = N_{\text{test}}$ equals the size of the test set) within the step range shown in the plots. The accuracy is on the full test set. The chosen steps have accuracy $\approx 1$ across tasks.

Equation 2 with $B = N_{\text{test}}$ given the specific length. We also include the full test results with $B = 1$ and $B = N_{\text{test}}$ from all test lengths in Figure 6, Appendix B.

**Convergence in Addition, Copy, Multiplication, and Unique Set.** We notice that for Addition, Copy, Multiplication, and Unique Set, the looped model somehow learns to converge for a certain number of steps after solving the task, even though we do not explicitly train the model to converge. The loss curves for these tasks are also smoother than those without convergence behaviors.

**The maximum confidence stopping criterion.** In Figure 5, the cross entropy loss (with $B = N_{\text{test}}$) reaches the lowest when the generalization performance is near perfect at the given test length, which indicates the maximum confidence rule chooses the right time to exit. We also show the full results in Figure 6 (Appendix B). The convergence nature of the tasks also affects the performance of the per-sample-based stopping criterion ($B = 1$) in Figure 6 (Appendix B): the per-sample stopping criterion performs almost as well as when using $B = N_{\text{test}}$ to decide for converging tasks, but the performance is worse in non-converging tasks.

## 7 LIMITATIONS AND CONCLUSION

Our current definition of $n$-RASP-L does not support tasks that require multiple loops followed by each other. Thus, one important future direction would be extending our current definition to support multiple loops and identifying a larger class of tasks that can be implemented only with multiple loops but not with a single loop. For training, direct looped training could be computationally demanding when the number of looped steps is too large. A possible workaround for more efficient training could be stopping the gradient tracking for earlier steps like Clark et al. (2023), but there might be a trade-off in performance and computation. We only train the looped Transformers for a limited number of steps and lengths due to a lack of computing resources. With more diverse training data, the looped model has the potential to generalize to even longer test lengths. We use NoPE for simplicity, and an orthogonal direction is to use more delicate positional embedding to further improve length generalization performance. Moreover, our step-dependent supervision requires the ground-truth number of steps in the training data, which is an additional requirement compared with normal end-to-end training. However, we still require fewer assumptions than CoT training.

In conclusion, we show that $n$-RASP-L problems can be learned by looped Transformer with step-dependent supervision on the final answer, and can be applied with an adaptive number of steps during inference time to improve generalization. Note that $n$-RASP-L, as a challenging algorithmic problem set, could cover more challenging reasoning problems than presented in the paper, and we believe that our method has the potential to generalize to more challenging tasks.

ACKNOWLEDGMENTS

The works is supported by NSF Award DMS-2023239, NSF CAREER Award CCF-2339978, Amazon Research Award, and a grant from FuriosaAI.

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

## A   $n$-RASP-L PROGRAMS

Here we provide the $n$-RASP-L programs for our Parity, Addition and Copy tasks in Listings 1, 2, 3. We also present the RASP-L library functions we use in Listing 4, which is partially taken from Zhou et al. (2024a).

```python
# Input example:  1 1 0 1 > # # #
# Output example: * * * * 1 # # #
# * indicates ignored token, > is EOQ, and # is EOS.

def parity_step(partial_ans_seq, seq):
    # align the last digit with the answer location
    seq = shift_right(seq, 1)
    # calculate XOR
    partial_ans_seq = (
        partial_ans_seq | seq) & (~(partial_ans_seq & seq)
    )
    return partial_ans_seq, seq

def parity_loop(seq, num_step):
    # get the question in the prompt
    prompt_mask = 1 - has_seen(seq, full(seq, EOQ))
    seq = mask(seq, prompt_mask)
    # init answer seq with 0
    partial_ans_seq = full(seq, 0)
    # generate EOS seq after EOQ
    end_seq = where(prompt_mask==1, full(seq, 0), full(seq, EOS))
    # perform parity steps
    for i in range(num_step):
        partial_ans_seq, seq = parity_step(partial_ans_seq, seq)
    # get answer with EOS
    ans_seq = partial_ans_seq
    end_seq = shift_right(end_seq, 1)
    ans_seq = where(end_seq==EOS, end_seq, ans_seq)
    return ans_seq
```

Listing 1: Parity.

```python
# Input example:  0 1 0 1 1 > # # # # # #
# Output example: * * * * * 0 1 0 1 1 # #
# * indicates ignored token, > is EOQ, and # is EOS.

def copy_step(seq, end_seq):
    seq = shift_right(seq, 1)
    end_seq = shift_right(end_seq, 1)
    return seq, end_seq

def copy_loop(seq, num_step):
    # generate EOS seq after EOQ
    end_mask = has_seen(seq, full(seq, EOQ))
    end_seq = where(end_mask==0, full(seq, 0), full(seq, EOS))
    # perform copy steps
    for i in range(num_step):
        seq, end_seq = copy_step(seq, end_seq)
    # get answer with EOS
    seq = where(end_seq==EOS, end_seq, seq)
    return seq
```

Listing 2: Copy.

```python
# Input example:  0 0 1 + 1 1 1 > # # # # # #
# Output example: * * * * * * * 1 0 0 0 # # #
# * indicates ignored token, > is EOQ, and # is EOS.

def addition_step(seq1, seq2, end_seq):
```

```python
        end_seq = shift_right(end_seq, 1)
        seq1 = np.array(seq1, dtype=bool)
        seq2 = np.array(seq2, dtype=bool)
        carry_on = seq1 & seq2
        # A XOR B = (A OR B) AND (NOT (A AND B))
        in_place = ((seq1 | seq2) & (~(seq1 & seq2)))
        in_place = shift_right(in_place, 1)
        seq1 = np.array(in_place, dtype=int)
        seq2 = np.array(carry_on, dtype=int)
        return seq1, seq2, end_seq

def addition_preprocess(seq):
        # generate EOS seq after EOQ
        end_mask = has_seen(seq, full(seq, EOQ))
        end_seq = where(end_mask==0, full(seq, 0), full(seq, EOS))
        # generate masks for the first and second summands
        seen_tok0 = has_seen(seq, full(seq, ADD_SIGN))
        seen_tok1 = has_seen(seq, full(seq, EOQ))
        mask1 = ~seen_tok0
        mask2 = seen_tok0 & (~seen_tok1)
        mask2 = mask2 & shift_right(mask2, 1)
        # get the first and second summands
        seq1 = mask(seq, mask1)
        seq2 = mask(seq, mask2)
        # align the first summand with the second
        induct_num1 = cumsum(mask1)
        induct_num2 = cumsum(mask2)
        target_index = firsts(induct_num1, induct_num2, default=0)
        seq1 = index_select(seq1, target_index)
        seq1 = mask(seq1, mask2)
        return seq1, seq2, end_seq

def addition_loop(seq, num_step):
        seq1, seq2, end_seq = addition_preprocess(seq)
        # perform addition steps
        for i in range(num_step):
            seq1, seq2, end_seq = addition_step(seq1, seq2, end_seq)
        # get answer with EOS
        ans = seq1
        ans = where(end_seq==EOS, end_seq, ans)
        return ans
```

Listing 3: Addition (in forward order).

```python
import numpy as np

def full(x, const):
        return np.full_like(x, const, dtype=int)

def indices(x):
        return np.arange(len(x), dtype=int)

def select(k, q, pred, causal=True):
        # compute attention matrix
        s = len(k)
        A = np.zeros((s, s), dtype=bool)
        for qi in range(s):
            for kj in (range(qi+1) if causal else range(s)):
            # k_index <= q_index if causal
                A[qi, kj] = pred(k[kj], q[qi])
        return A

def sel_width(A):
        return np.dot(A, np.ones(len(A))).astype(int)
```

```python
def aggr_mean(A, v, default=0):
    out = np.dot(A, v)
    norm = sel_width(A)
    out = np.divide(
        out, norm, out=np.full_like(v, default,dtype=float),
        where=(norm != 0)
    )
    return out.astype(int)

def kqv(k, q, v, pred, default=0, reduction='mean'):
    return aggr_mean(
        select(k, q, pred), v, default=default,
        reduction=reduction
    )

def seq_map(x , y, func):
    # tokenwise map over two sequences
    return np.array([func(xi, yi) for xi, yi in zip(x,y)])

def shift_right(x, n, default=0):
    # shifts sequence x to the right by n positions
    # (other positions filled with default)
    return kqv(indices(x)+n, indices(x), x, equals, default=default)

def where(condition, x_if, y_else):
    # equivalent to np.where(condition, x_if, y_else)
    x_masked = seq_map(x_if, condition, lambda x, m: x if m else 0)
    y_masked = seq_map(y_else, condition, lambda y, m: y if not m else 0)
    return seq_map(x_masked, y_masked, lambda x, y: x if y == 0 else y)

def has_seen(x, queries):
    return kqv(x, queries, full(x, 1), equals, default=0)

def firsts(x, queries, default=-1):
    # find the index of the first occurrence of each query[i] in x
    return kqv(
        x, queries, indices(x), equals,
        default=default, reduction='min'
    )

def mask(x, bool_mask, mask_val=0):
    # equivalent to x*bool_mask + default*(~bool_mask)
    return where(bool_mask, x, full(x, mask_val))
```

Listing 4: Library functions from Zhou et al. (2024a).

## B  RESULTS FROM THE ADAPTIVE STOPPING CRITERIA

Here we present the length generalization results in Fugire 6 with different stopping criteria: 1) Oracle 2) Maximum confidence based on the full test set using Equation (2) with $B = N_{\text{test}}$ given the specific test length and 3) Maxumum confidence using Equation (2) where $B = 1$ is calculated per sample (we still calculate the accuracy across the full test set). For tasks with converging behaviors (Addition, Copy, Multiplication, Unique Set), per sampler stopping criterion works almost as well as using $B = N_{\text{test}}$, while for tasks without the converging behavior (Parity and Binary Sum), per sampler stopping criterion does not work as well as using $B = N_{\text{test}}$.

Besides, we also visualize another baseline which is using FOP with looped structure but with a fixed number of steps, which is denoted as "FOP-Loop-Fixed" in Figure 6, showing that using variable depth generally outperforms fixing the depth of the model.

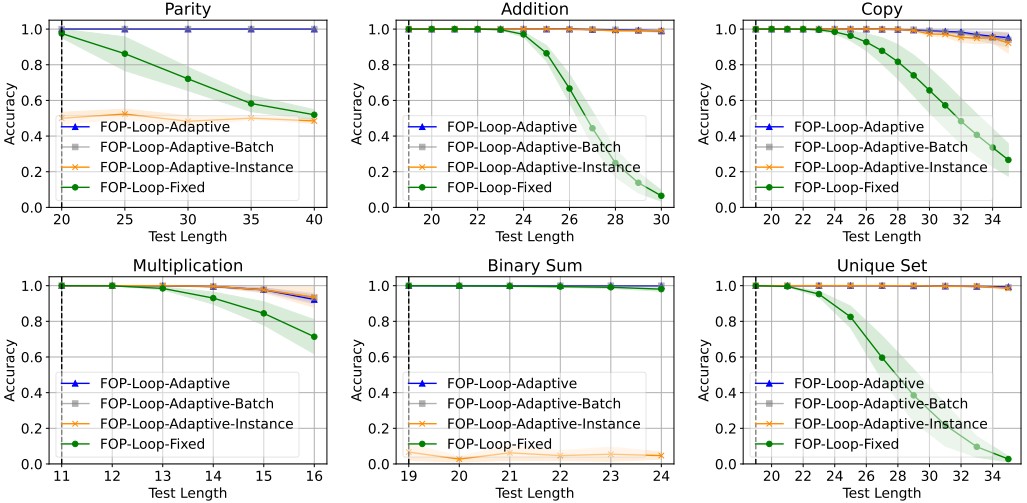

Figure 6: Effect of different stopping criteria. "FOP-Looped-Adaptive" indicates stopping with the pre-defined number of steps; "FOP-Looped-Adaptive-Batch" indicates stopping with Equation (2) where $B$ is the size of the test set; "FOP-Looped-Adaptive-Instance" indicates the number of steps is chosen per sample, with Equation (2) where $B = 1$ (the accuracy is still calculated across the dataset).

## C ABLATION STUDY

We present the generalization performances of our method compared with FOP variants in Figure 7 as mentioned in Section 6.3.

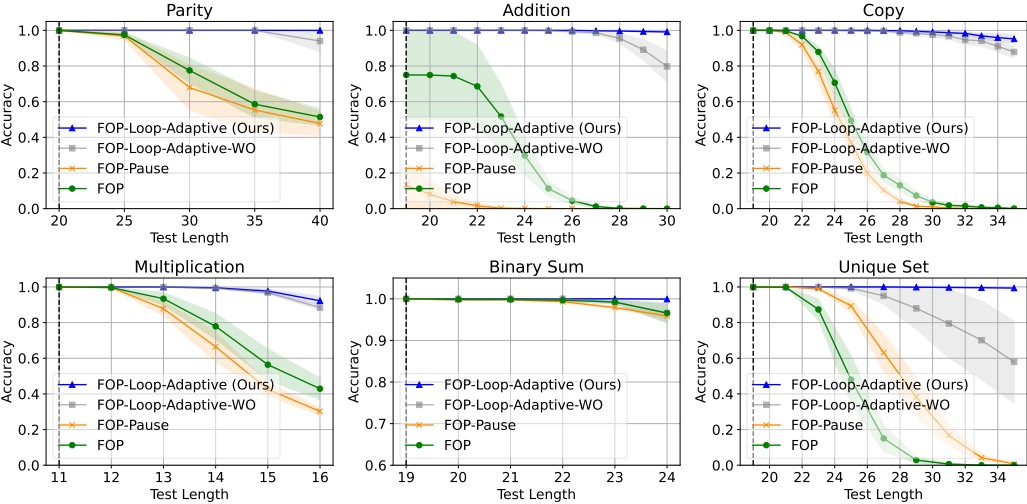

Figure 7: **Ablation study**. Our looped Transformer model with adaptive depth generalized better than FOP variants across studied tasks, including the variant of our method without input injection, and FOP with pause tokens. The vertical dashed line indicates the maximum training length.

## D VISUALIZATION OF USING PAUSE TOKENS IN NTP AND FOP

We visualize NTP-Pause in Figure 8 and FOP-Pause in Figure 9 respectively, where we add a fixed number of pause tokens (3 in the figures, 20 in our experiments) before outputting the final answer during both training and inference.

Table 2: Task-specific experimental hyperparameters. "Number of Heads" and "Block Depth" define the size of the looped decoder block. "Interval" denotes the number of training steps between successive increases in the input sequence length.

| Task | Number of Heads | Block Depth | Interval |
|---|---|---|---|
| Parity | 64 | 1 | 500 |
| Copy | 8 | 2 | 1000 |
| Addition | 8 | 3 | 1600 |
| Multiplication | 8 | 4 | 500 |
| Binary Sum | 16 | 2 | 500 |
| Unique Set | 8 | 3 | 1000 |

## E  INFERENCE TIME COMPLEXITY

Here we present the inference time complexity for our method, vanilla NTP and vanilla FOP.

Assume that the maximum length of the training set is $n$, the number of steps needed is $T(n)$, and the number of layers in each step is $k$. Assume that NTP and FOP are using a fixed number of layers $C$. And we test on length $n'$.

For the first stopping criterion where we know $a(n')$, our inference time would be $O(ka(n')n'^2)$, and NTP (with KV cache) and FOP will be $O(Cn'^2)$. For the second criterion, we need to specify the maximum number of steps in order to find the step with maximum confidence. So our inference time would be $O(kN'n'^2)$, where $N'$ is the maximum number of steps.

In NTP and FOP, we use some $C \approx kT(n)$ in our experiments such that they use similar compute during training. Our inference time is then slightly longer than NTP with KV cache and FOP since we use more steps than the fixed-depth models.

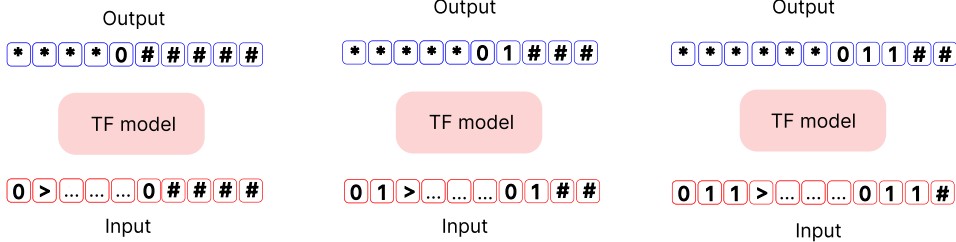

Figure 8: NTP-Pause visualization. Examples are from the Copy task. "..." indicates the pause token.

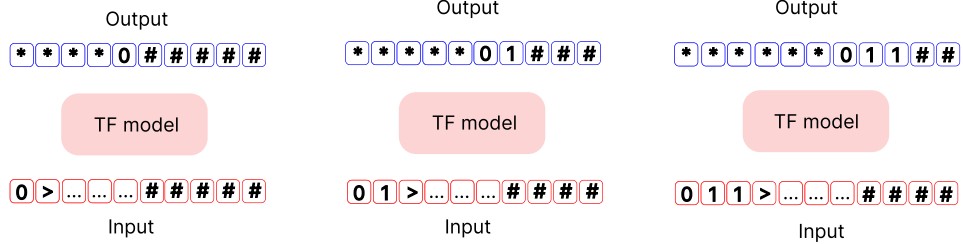

Figure 9: FOP-Pause visualization. Examples are from the Copy task. "..." indicates the pause token.

## F  EXPERIMENTAL DETAILS

**Model architecture.**  We use the decoder-only GPT-2 model with NoPE and 256 embedding dimensions as the basic block for the looped iterations with task specific configuration in Table 2. We first convert the input to the embedding space, perform the looped computation in the embedding space, and then decode the final output. For input injection, we use a similar technique as Yang et al.

(2024) that adds the original input embedding to each iteration as part of the input. For vanilla NTP, we adopt the same training scheme, but trained with autoregressive loss instead. For NTP-Pause and FOP-Pause, we add 20 pause tokens before outputting the final answer. For FOP-Looped-Fixed, we fixed the model depth to be the maximum depth used in FOP-Looped-Adaptive.

**Training and evaluation details.** For the training distribution, we adopt the online training scheme following Zhou et al. (2024a) where each batch is i.i.d. sampled. Given any length, the probability of each possible character is evenly distributed instead of from a finite train set to avoid over-fitting, and the length is also evenly distributed. We also use a curriculum to gradually increase the maximum training length (see Table 2 for the specific setup for each task). We use AdamW optimizer and decay the learning rate from $10^{-4}$ to 0 with cosine decay schedulers after reaching the maximum training length with batch size 64, and train for a total number of 100k gradient steps. Additionally, non-converging tasks are less tolerant when choosing which step to stop, and we find using the exponential moving average (EMA) of model parameters helpful for Parity and Binary Sum with a factor 0.9999. For evaluation, we test with $6400$ random samples in Figure 4, 5, 7, 6, and report the mean exact match accuracy and standard error from five training runs with different random seeds.

