# OpenReview forum: "Looped Transformers for Length Generalization"
_ICLR.cc/2025/Conference — ICLR 2025 Poster_

### Official Review · Reviewer_Fhkc · 2024-10-24

**Soundness:** 4
**Presentation:** 3
**Contribution:** 2
**Rating:** 6
**Confidence:** 4

**Summary:**

Empirically explores the ability of looped Transformers, i.e. Transformers that repeatedly apply the same block of layers, to length-generalize on several algorithmic tasks, including copy, parity, and addition. First, the authors manually derive length-generalizing solutions to the considered tasks through a variant of the RASP language, which they term n-RASP-L. Then, based on these ground truth solutions, they show that looped Transformers length-generalize well when trained with access to the true number of steps required to compute the output for a given input.

**Strengths:**

1. The paper is mostly well-written and easy to follow.

2. Demonstrates that, given knowledge of the number of steps required to perform a given task, a certain looped Transformer, which jointly predicts the full output sequence, tends to learn a length-generalizing solution. The length-generalizing capabilities of this looped Transformer are shown to surpass baselines that use next-token prediction.

**Weaknesses:**

1. The main weakness of the current paper is that the significance of the results is somewhat limited. In particular, I find that it falls somewhere in between works that are practically relevant and those that may not be practically relevant, but improve our understanding of certain phenomena. On the one hand, this work shows empirically that, in some algorithmic tasks for which we already know how to write explicitly a length-generalizing solution (in terms of looped Transformer weights), looped Transformers generalize well to longer lengths, if they have access during training to the number of steps required for solving the task for a given input. Consequently, the practical relevance is limited since the proposed method requires that we already know how to manually write a length-generalizing solution, in which case there is arguably no point in learning. On the other hand, this work does not provide much in terms of understanding why or how looped Transformers are able to length-generalize.

    Note that one may consider the demonstration of such length generalization to be possible as a main contribution. Yet, the ability to extrapolate through recurrence of layers has been demonstrated in the past, albeit for other architectures (see Bansal et al. 2022 [1], which notably do not require knowing the ground truth number of steps in training).

2. A related issue is the usage of ground truth stopping time during inference. The quantities reported in Figure 5 seem to be for a single training example, yet it is not entirely clear. If so, then how does the maximum confidence stopping criterion fair across the dataset? It would be useful to report results similar to those of Figure 4 but when using the proposed stopping criterion as opposed to the ground truth stopping time, which should be unknown.

Overall, my assessment of the paper tends towards the positive side, yet it is not a clear accept due to the substantial limitations mentioned above. Specifically, the significance of the contributions can be greatly improved if it would be possible to remove the dependence on knowing the ground truth number of steps required to solve the task for a given input during training (and by how it seems from the current results, during test time as well).


Additional (more minor) comments:
- In Definition 3.1, it seems that the intention is for $P’$ to be some RASP-L program, as opposed to just a program. Otherwise, trivially any program $P$ is an n-RASP-L program by choosing $P’ = P$ and $T(n) = 1$.
- In Equation (2), I believe that the criterion should be an argmin over the cross entropy loss instead of an argmax.


[1] Bansal, A., Schwarzschild, A., Borgnia, E., Emam, Z., Huang, F., Goldblum, M., & Goldstein, T. (2022). End-to-end algorithm synthesis with recurrent networks: Extrapolation without overthinking. Advances in Neural Information Processing Systems, 35, 20232-20242.

**Questions:**

1. Are the quantities reported in Figure 5 indeed for a single training example? When using the maximum confidence criterion, how do the results compare to the ones reported in Figure 4 with access to the ground truth number of steps?

2. In Bansal et al. 2022, they avoid the need for knowing the exact number of steps during training and inference. Have you tried using similar heuristics?

---

> ### Author Response · Authors · 2024-11-21
> **Rebuttal**
>
> Thank you for the acknowledgment of our work and the detailed comments. We respond to each point in the following:
>
> ### About limited significance, and comparison to Bansal et al. [1]
>
> Thank you for the detailed comments! We acknowledge that generalizing to larger images could share certain similarities to generalizing to longer input sequences. However, we would like to emphasize that although similar extrapolation with recurrent networks has been studied in [1] (which uses recurrent convolutional layers instead of Transformers), our work still has significant contributions: 1) We provide the theoretical n-RASP-L framework to motivate why such recurrence would help length generalization, based on specific domain knowledge in Transformers  2) We design Transformer architectures and the training setup according to the proposed theoretical framework and 3) We show empirical success of the proposed method. Showing looped Transformers could help length generalization in our work still requires significant effort. given that the theoretical framework is highly different.
>
> As for requiring the number of predefined steps, we believe that this is a relatively weaker form of side information, compared to the other side information used in other methods such as scratchpad training which is a common and popular approach in the length generalization domain.
>
> Besides, in our NTP-Loop baseline, we used a fixed number of loops to do the next token prediction as a baseline, which does not require knowing the number of steps needed. Our experiments show that certain extra side information like # of predefined steps would help the model avoid learning shortcut solutions that fail at length generalization. However, further relaxing the requirements could be interesting for future work.
>
> Again, we thank you for extending the discussion on the comparison with Bansal et al. [1], and would incorporate a more comprehensive discussion than the current introduction and the related work section about this work.
>
> [1]  End-to-end algorithm synthesis with recurrent networks: Extrapolation without overthinking
>
> ### About the stopping criterion and confusion about Figure 5
>
> We want to clarify that the quantities reported in Figure 5 are from the test examples with 100 times the training batch size, which is introduced in the experimental details in Appendix F in our draft (there is no training example). In Figure 5 the pre-defined number of steps is unknown and we just run a maximum number of steps and then select the stopping time based on the confidence of the output.
>
>
> ### Minor issues
>
> Thank you for pointing them out! We will fix them in the revised draft.

---

> > ### Comment · Reviewer_Fhkc · 2024-11-21
> >
> > Thank you for the response, I have read it and the other reviews carefully. I have a couple of follow up remarks.
> >
> > 1. The response has not addressed my concern regarding the necessity of knowing the stopping time, especially during inference (see question #1 in the original review). In particular, when using this stopping criterion it is not clear from the paper nor the response how the results compare to the ones reported in Figure 4, in which access to the ground truth number of steps is assumed. If indeed, as claimed in Section 6.4, the stopping criterion chooses the correct step near-perfectly, then why not report the full results (i.e. results analogous to Figure 4) when using this criterion? I believe it is worth being more transparent here. To be clear, I do not believe that if the stopping criterion is suboptimal, then this is a blocker for publication. Rather, more concerning is the opaqueness regarding to how well it works.
> >
> > 2. I would recommend the authors to specify that the quantities in Figure 5 are means across the test examples in the main text, as opposed to an appendix. What is still not clear to me though is what does the vertical line marking the chosen step correspond to in Figure 5? Isn't the stopping criterion different for each example, or is it chosen once for the whole test set?

---

> > > ### Author Response · Authors · 2024-11-24
> > >
> > > Thank you so much for the careful review and the suggestions! We acknowledge that the original description needs to be clearer for Figure 5 and update the description in Section 4.2 with new results in Figure 9, Appendix D to clarify the ambiguity. Please check the draft with updates marked in blue.
> > >
> > > ===
> > >
> > > **1. “Please report the full results when the stopping criterion” (paraphrased)**
> > >
> > > Our response: As you suggested, we reran experiments with our adaptive stopping criterion applied and presented the result in Figure 9, Appendix D as suggested. (We are happy to move this to the main body if the reviewers think that’s a better idea – for now, we couldn’t do so due to the page limit.)
> > >
> > > **2. “Stopping criterion is applied differently for each example or is it chosen once for the whole test set?” (paraphrased)**
> > >
> > > Our response: The stopping criterion visualized in Figure 5 was chosen for the whole test set. (That’s why we had one figure per task, and one vertical line representing the optimal stopping time per task.)
> > >
> > > However, inspired by your question, we also tested the same stopping criterion “per-sample.” We found that for the converging tasks, both per-sample and per-test-set have similar performance, but for non-converging tasks, the per-sample stopping criterion does not perform well (even for in-distribution). We believe that it is because converging tasks are more tolerant with respect to when to stop, while non-converging tasks are not, as shown in Figure 5.
> > >
> > > Accordingly, we updated Equation (2) to include an extra hyper-parameter $B$, denoting the number of test samples used to decide when to stop. Using this, we now have results for both $B = 1$ and $B = n_{test}$.
> > >
> > > ===
> > >
> > > We hope this addressed your questions/concerns. Please let us know if you have any additional questions or suggestions. Again, we really appreciate your thoughtful comment!

---

> > > > ### Comment · Reviewer_Fhkc · 2024-11-24
> > > >
> > > > Thank you for providing additional details and experiments regarding the stopping criterion. The updates made to the paper have fully addressed both of the above-mentioned concerns. I therefore would like to maintain my initial positive assessment of the paper.

---

### Official Review · Reviewer_imEP · 2024-11-03

**Soundness:** 3
**Presentation:** 3
**Contribution:** 3
**Rating:** 8
**Confidence:** 4

**Summary:**

This paper examines how looped transformers perform in terms of length generalization. The focus is on n-RASP-L problems, which are problems that can be tackled using a loop of a single RASP-L program. The concept is that Transformers can learn steps that are independent of length, employing a flexible number of iterations in the looped transformer to achieve length generalization. The authors first demonstrate that n-digit addition, n-bit parity, and copying n symbols can be addressed with n-RASP-L solutions. They then reveal that when utilizing the looped transformer with adaptive stopping time, the results exhibit significantly stronger length generalization compared to next token prediction (NTP) and other methods like using pause tokens or NTP-loop with a fixed stopping time.

**Strengths:**

Overall, I really liked the paper, I think that using a looped transformer to achieve length generalization is an interesting idea that was not studied in the past to my knowledge. This paper complements all the other techniques (universal transformers, different types of position emebedding, etc.) that were used in the past for length generalization The paper is well-written and well-explained. This is why I advocate for acceptance of this paper.

**Weaknesses:**

I would like to raise the following weaknesses/questions regarding this paper:

- **Lack of other baselines**: What would happen if you have a very deep universal transformer? Universal transformers also have shared parameters and looks equivalent to the loop transformer. The depth may play the role of the number of loops. Would this be equivalent to the fixed loop NTP? It would be interesting to run the same experiments with a universal transformer.

- **Comparison with other methods**: Where would you position the looped transformers in the list of all the tricks for length generalization? Are the effects similar or complementary to change of the input (index hinting, reverse order of operands, etc.) ? Changes of positional encoding? Chain of Thought? It would be interesting to understand this by making combinations of the tricks with looped transformers with other tricks and analyze the performance differences.

-  What is the depth of the encoder block in the loop transformer? I think this information is important to put in the main paper.

- **Adaptive inference time**: I think one weak point of the method is actually coming up with an adaptive inference time. The methods that are proposed are nice but may look a bit hacky. Do you think one could learn this adaptive inference time?

- In Figure 2, which adaptive inference time method is used for FAP-Loop-Adaptive?

- Lastly, this is a wild question: have you tried your method on problems where there is no n-RASP-L solutions?  Would it still work better than just doing NTP?

**Questions:**

I listed my questions in the weaknesses section.

---

> ### Author Response · Authors · 2024-11-21
> **Rebuttal**
>
> Thank you for acknowledging our work and providing detailed comments. We respond to each point in the following:
>
> ### About Universal Transformers
>
> Our architecture is indeed similar to Universal Transformers (UT), but not equivalent. As discussed in Table 1 in our draft, there are certain design choices and architectural differences compared to UT, which is tailored to our n-RASP-L setup. Notice that an apple-to-apple comparison to the original Universal TF is not trivial due to these differences. The most similar baseline we have is the fixed loop NTP (NTP-Loop), but it is still not exactly the same as UT.
>
> ### About other comparisons
>
> Thank you for the suggestions! For this paper, we want to study whether the proposed architecture and training could help length generalization, and stick to NoPE to avoid the effect of different positional encodings (positional encodings could not be represented by RASP-L). This is orthogonal to other tricks like positional embeddings, index hints, and other format changes. Also, most of the format designs are for NTP but not for FAP. However, it would indeed be interesting to change the positional encoding in our current method to see whether the performance could be further improved, and we plan to add such experiments in the final draft.
>
> ### Depth of the encoder block
>
> Just to clarify, we do not have encoder blocks and only use decoder blocks in our paper since RASP-L is for causal models.
>
> ### Adaptive inference time
>
> There are certain halting mechanisms in Universal Transformers and Ponder Net (as discussed in Table 1 in our draft) that actually learn some weights about adaptive inference time. Our approach is more compatible with our n-RASP-L formulation, and exploring halting techniques could be interesting for future work.

---

> > ### Comment · Reviewer_imEP · 2024-11-23
> >
> > I thank the authors for their rebuttal. After reading the other reviews, I still believe that the paper is interesting and maintain my score.

---

> > > ### Author Response · Authors · 2024-11-24
> > >
> > > Thank you for your response. We are happy to hear that you’re willing to support our work.
> > >
> > > We just uploaded a revised draft with all the reviewers’ comments incorporated. Please let us know if you have any further questions.

---

### Official Review · Reviewer_3KX9 · 2024-11-04

**Soundness:** 3
**Presentation:** 3
**Contribution:** 3
**Rating:** 6
**Confidence:** 3

**Summary:**

This paper investigates the length generalization problem of Transformer models, which refers to the inability of the model to deal with longer samples than encountered during the training phase. While recent literature has focused on modifying the positional embeddings and the input formats, this paper proposes to use Looped Transformers, which can dynamically adjust their computation steps according to the problem length. The authors define n-RASP-L problems to figure out which problems can be solved by Looped Transformers. Then, they train the models on these tasks (parity, copy, binary addition, binary sum, binary multiplication, unique set) under a full-answer prediction setup. Empirically, the trained models could successfully length-generalize to longer lengths by appropriately adapting the number of loops at inference time.

**Strengths:**

- The paper is well-structured and clearly written.
- The introduction of Looped Transformers is well-motivated and effectively argued.
- The results are strong and solid. They do not require the use of a scratchpad. Also, the prediction is conducted using an end-to-end, full-answer prediction setup, which is a more general way than the conventional next-token prediction setup.
- The paper clearly illustrates that the model can determine the number of steps to take on its own and does not require T(n) in the test time.

**Weaknesses:**

Weakness 1: Applicability Limited to n-RASP-L Tasks

- The approach is limited to tasks that belong to n-RASP-L categories, as it requires the ground-truth number of steps in the training data.

Weakness 2: Insufficient Experimentation.

- ***Effect of Curriculum Learning.*** How does the model perform without curriculum learning? Is the use of curriculum learning necessary?

- ***Tolerance to Step Counts.*** I am curious whether this method will still perform well with different choices of T(n). For example, for tasks like parity, would the model maintain its performance if T(n) were set to n+1 rather than n? What about 2n instead of n? This question stems from the possibility that there might be more efficient solutions to n-RASP-L problems than human-designed ones, which could work with fewer steps. Testing whether the model is robust under overestimated T(n) values could help verify the robustness of this approach.

- Overall, the paper requires more ablation studies.

**Questions:**

Q1. In Figure 5, why do some tasks perform well even when exceeding the step count, while others degrade immediately? For instance, the performance of the parity task and the binary sum task immediately drops when executed with additional steps, whereas the addition, multiplication, and copy tasks retain accuracy to some extent.
- Particularly for the copy task, the selected step count is significantly higher than the actual number of steps required, which seems unusual to me.

Q2. Are there any tasks whose T(n) is nonlinear (e.g. sqrt(n), n^2) to the length of the input sequence? It would be interesting to see experimental results for such tasks.

Q3. Why is the output reversed for binary multiplication (but not for binary addition)?

---

> ### Author Response · Authors · 2024-11-21
> **Rebuttal**
>
> Thank you for acknowledging our work and providing detailed comments. We respond to each point in the following:
>
> ### Limitation of n-RASP-L tasks
>
> We acknowledge that our scope is learning to solve n-RASP-L tasks while utilizing the number of predefined steps during training.  our experiments show that using such extra side information helps the model avoid learning shortcut solutions that does not length generalize. We also believe that this is a relatively weaker form of side information, compared to the other side information used in other methods such as scratchpad training. Our work could be viewed as a first attempt in this domain which uses test time scaling in terms of increasing the number of looped steps in Transformers, and potentially relaxing the assumptions could be interesting future work.
>
> ### Effect of curriculum learning
>
> As shown in [1], curriculum learning does not affect the final performance significantly. It is pretty standard in training Transformers with increasing lengths like [1] and we find it helps speed up the training so we use it as a common trick for all methods we compare. We will add a remark in the experimental section in the revised draft.
>
> [1] What Can Transformers Learn In-Context? A Case Study of Simple Function Classes
>
> ### “Tolerance to step counts”
>
> We conduct experiments in Parity with 1 and 2 additional steps in training and observe a decay in the performance of longer-length generalization with length 50. The performance in length 30 remains near optimal which shows some extent of robustness. As for more efficient solutions, there might be solutions with fewer steps, but might not just be a constant shift in terms of the number of steps.
>
> | Model/Test Length | 10  | 30  | 50      |
> |-------------------|-----|-----|---------|
> | Original          | 1.0 | 1.0 | 1.0     |
> | 1 more step       | 1.0 | 1.0 | 0.96875 |
> | 2 more steps      | 1.0 | 1.0 | 0.375   |
>
>
> ### Question about converging/non-converging behaviors
>
> About the performance after the number of steps we have in the solutions, there are two cases as discussed in Section 6.4: converging, or not converging. For the tasks with converging behaviors, it might appear that even after the number of steps predefined in our solution, it would still maintain the performance like the copy task. Our conjecture is that during training, the model learns some kind of index hint and refines the output in place in each step (instead of shifting the output location in each step as our n-RASP-L solution), so it tends to find some fixed-point solution with more steps (However, there might also be other n-RASP-L solutions for copy that are different from our solution and potentially with a similar number of steps needed). For other tasks like parity, the model learns something very similar to our n-RASP-L solution and only outputs the right answer at the pre-defined number of steps. We think it is still an interesting open question, and whether the model learns such kind of behavior might be task-dependent. We will add a detailed discussion in the final draft.
>
> ### Nonlinear T(n)
>
> There might also be solutions for parity with log(n) steps as Reviewer gBs4 pointed out. Further exploring such tasks could be interesting for future work.
>
> ### Reversed vs not reversed
>
> For binary addition, we found an n-RASP-L solution without reversed inputs so we want to see whether our training can achieve this given that most works focus on the reversed output in binary addition and fail on non-reversed output format. We are not sure whether there exist n-RASP-L solutions for multiplication without reversed output, so we stick to the common setup with reversed output, especially since recent work has shown that the reversed format also makes multiplication easy to learn [2].
>
> [2] Positional Description Matters for Transformers Arithmetic

---

> > ### Comment · Reviewer_3KX9 · 2024-11-23
> >
> > Thank you for the response. However, as reviewer gBs4 mentioned, I would like to see the revised paper as ICLR conference allows authors to revise their paper during the rebuttal period.
> >
> > Overall, while the applicability of the proposed approach is limited, I find the paper intriguing and believe that this paper offers an interesting direction in the iterature on length generalization. I will maintain my score.

---

> > > ### Author Response · Authors · 2024-11-24
> > >
> > > Thank you for your response. We are happy to hear that you’re willing to support our work.
> > >
> > > We just uploaded a revised draft with all the reviewers’ comments incorporated. Please let us know if you have any further questions.

---

### Official Review · Reviewer_gBs4 · 2024-11-04

**Soundness:** 4
**Presentation:** 4
**Contribution:** 3
**Rating:** 6
**Confidence:** 4

**Summary:**

- This work studies the efficacy of Looped Transformers for Length Generalization of several algorithmic tasks whose computation complexity is known (as a function of the query length).
- The paper proposes the definition of $n$-RASP-L, a generalization of the RASP-L computation model allowing the loop of RASP-L programs. It is shown, under a general framework called full-answer prediction (FAP), that some tasks (Copying binary sequence (allowing duplicates), Parity, and Binary Addition) have their own $n$-RASP-L program with a linear number of steps in problem length.
- The authors propose training Looped Transformers (with input injection and curriculum learning) to learn $n$-RASP-L-programmable tasks, where the ground-truth number of steps is known for each task during training. They also propose two variants of inference methods: either we retain the knowledge about the number of steps at inference time (*Oracle*), or we adaptively decide the number of iterations based on the confidence of FAP (*Maximum confidence*).
- The proposed method is tested on several algorithmic tasks.

**Strengths:**

S1. The paper is written and organized well. Overall, the presentation of the methodology and empirical results is clear and easy to follow.

S2. The idea behind the proposed method is neat and plausible. It is natural to think about adaptively scaling the depth of the model according to the problem length or the problem complexity. This paper successfully implements this idea to solve various interesting algorithmic tasks with the power of Looped Transformers. Also, $n$-RASP-L is an interesting but intuitive generalization of the RASP-L framework by allowing the loops.

S3. The proposed answer-generation framework called FAP is also an interesting component of this work. It might be of separate interest to study.

S4. The paper presents extensive ablation studies on several components of the proposed method. Also, the empirical results (length generalization performances) are impressive enough to convince the readers about the proposed method’s efficacy.

**Weaknesses:**

**W1. The definition of $n$-RASP-L (Definition 3.1) can be improved.**

- I think the equation “$T(n): \mathbb{N} \rightarrow \mathbb{N}$” should be corrected to “$T: \mathbb{N} \rightarrow \mathbb{N}$” because $T$ (instead of $T(n)$) is a function of input length $n$ representing the number of steps inside a task-solving $n$-RASP-L program.
- In (2), I guess $P’$ should be a RASP-L program, which is unspecified in the definition.
- Should $P$ be decomposed to a sequential application of $P’$, i.e., $P = (P’)^{T(n)}$? I don’t think this is exactly true because there are pre-/post-processing parts inside the proposed $n$-RASP-L programs (in Appendix A). Can the same RASP-L program $P’$ handle such parts? (It might be true because of the experimental results, but I cannot fully understand this part.) If not, I guess the definition should be modified to include the pre-/post-processing parts. For example, $P = P_{\tt pre} \circ (P’)^{T(n)} \circ P_{\tt post}$.

**W2. “Ground truth” number of steps?**

- According to Definition 3.1, a program $P$ suffices to be an $n$-RASP-L if a corresponding $T(n)$ exists. Indeed, Propositions 3.2, 3.3, and 3.4 claim and prove the existence of $T(n)$ for the Parity, Copy (with duplicates), and Binary Addition tasks, respectively.
- My question is about the uniqueness or optimality of such $T(n)$’s. There might be a clever way to construct another RASP-L program $\tilde{P}$ so that $P$ can be implemented with $\tilde{T}(n)$ steps of applying $\tilde{P}$, where $\tilde{T}(n)$ is much smaller than the previously known $T(n)$ (e.g., $\tilde{T}(n) \in o(T(n))$). It may happen since there is no uniqueness guarantee or lower bound result on $T(n)$.
    - If I venture a guess, I would say it might be possible to implement an $O(\log n)$-step $n$-RASP-L solution for the Parity task by using the parallelism of the transformer architecture. Please correct me if I am wrong. Also, I understand if it is impossible to show whether this bold guess is true. If you are interested, there are some (probably) useful references about logarithmic-depth transformers [1,2].
- However, the authors keep using the phrase “ground truth number of steps” throughout the paper, which may lead to misunderstanding that the only way to implement the given $n$-RASP-L program is by using a loop of length $T(n)$.
- If two different $T(n)$’s can be applied to a single $n$-RASP-L-programmable task, it might be interesting to observe whether the model’s performance changes depending on the choice of $T(n)$.
- Furthermore, if multiple choices of $T(n)$’s exist for a given task, does knowing only one of them suffice to train reasonably performant Looped Transformers? If we know more than one, how should we choose $T(n)$ when we train the model?

**W3. Shouldn’t we consider the input injection when implementing an $n$-RASP-L program for the given task?**

- The input injection seems to be an important component of their experiments. Since it changes the input vectors of each layer, I guess the task-solving algorithm under input injection might be different from that without it.
- However, I can’t see that the $n$-RASP-L programs provided in Appendix A reflect the input injection. As I inspect inside the loop of each program, every iteration only reuses the calculation(s) from the previous iteration right before the current one.
- Shouldn’t we consider the very first input sequence and the result from the previous iteration when implementing the loops? Or is it a valid implementation of input injection? Getting even further, Is there any way to embed the input injection into the $n$-RASP-L programs?

**W4. The proposed training method requires prior knowledge of the task’s structure.**

- The proposed method is limited in that it requires a prior understanding of the structure (e.g., $T(n)$) of the task where we want to train a model. This is because it hinders fully end-to-end training.
- Are Looped Transformers still useful for achieving length generalization even when we don’t (or cannot) know the exact expression of $T(n)$?
- Besides, it seems that the depth of the decoder block is determined based on the complexity/difficulty of the subroutine $P’$ at each step inside the loop (Appendix F). How are they actually chosen? Or, how should we decide the size of the repeating decoder block?

**W5. Some experimental details seem missing or wrong.**

- I guess Equation (2) has a typo: shouldn’t it be arg-main instead of arg-max?
- In Binary Addition, it seems that $T$ is chosen to be $n$ (the length of each operand). However, Proposition 3.4 claims that $T(n)=n+1$ for the same task. Why is there a discrepancy between theory and experiment?
- In Binary Multiplication, I guess some words are used in a wrong way. In Lines 417-418, I think it should be: “We define the problem length to be the **length** of the second **number**, and set $T$ to be the product of the lengths of two **numbers**.”
- In Section 6.1.2, are input injections also applied to NTP-based methods? Also, I’m not sure why it is fair to compare their method (based on FAP) to NTP methods with the architectural setting “…with a depth 20 times the depth of the looped block” because such depth might be suboptimal for NTP-based methods.
- Although the paper indirectly showcases that their adaptive decision of the number of steps works quite well via Figure 5, it would be better to display similar performance plots to Figure 4 (plots based on the “Oracle” inference) but using the adaptive confidence-based method instead, at least in their appendix.

**W6. Minor writing issues**

- Section 4.1, fourth bullet point: I guess $T(n) \in \\{T(1), \ldots, T(n_{\rm max})\\}$ is correct ($T(1)$ instead of $1$).
- Equations (1) and (2) have several weird-looking brackets (too many open brackets etc.)
- Line 510: Use *less* abbreviations like “w.r.t.”

---

**References**

[1] Sanford, Clayton, et al. "Transformers, parallel computation, and logarithmic depth." ICML 2024.

[2] Sanford, Clayton, et al. "Understanding transformer reasoning capabilities via graph algorithms." NeurIPS 2024.

**Questions:**

**Q1. Question on the visualization in Figure 3**

- Why don’t the illustrations in the figure contain any “#” (EOS) tokens? Is it due to the pre-processing?

**Q2. Do the trained Looped Transformers simulate the $n$-RASP-L program?**

- Although it might be difficult to reverse-engineer a trained transformer model to figure out what algorithm it actually simulates or implements, it might be interesting if we can observe any kind of similarity between it and the $n$-RASP-L program.

---

> ### Author Response · Authors · 2024-11-21
> **Rebuttal (1/2)**
>
> Thank you for acknowledging our work and providing detailed comments. We respond to each point in the following:
>
> ### The definition of RASP-L:
> Thank you for pointing them out! We acknowledge all proposed suggestions and will incorporate them into the revised draft.
>
> ### “Ground truth” number of steps
> Thank you for the comments and for providing the references on the logarithmic-depth transformers! We acknowledge that we only provide one possible n-RASP-L solution in the draft, and we did not mean that it is the only solution. We will use “pre-defined number of steps” to replace the term “ground truth number of steps” in the revised draft.
> Besides, it is an interesting question on which to choose if there are multiple n-RASP-L solutions with different T(n) functions. Some solutions might be easier to learn while others could be harder. Thus, the generalization performance might depend on many factors: the training data distribution, the architecture of the looped layers, etc. There might be no general criterion for the choice. In practice, we could use a validation set to choose T(n) with the best validation accuracy.
>
> ### About input injection
> Thank you for your question. In fact, a single input injection operation could also be embedded in an RASP-L program: Notice that identical mapping could be represented by RASP-L. We can express input injection if we double the size of the embedding space in half and let some MLP layer add the outcomes from two embedding spaces. However, when designing the n-RASP-L solutions for specific programs, we did not find it necessary to add input injection operations, while we found it more useful in practice. We believe that this is because adding explicit input injection helps gradients flow better in looped models, which is a common trick as discussed in [1][2][3]. We also provided the performance with and without input injection in Figure 6, Appendix B, where there is a slight decay in the test accuracy without input injection, but still outperforming other baseline methods.
>
> [1] Deep equilibrium models
> [2] Looped transformers are better at learning learning algorithms
> [3] End-to-end algorithm synthesis with recurrent networks: Extrapolation without overthinking
>
> ### Prior knowledge of the task structure
> Thank you for the comments. First, we tested (although under the NTP scheme) using a fixed number of loops for all problems in Figure 4 “NTP-Loop” baseline, where we fixed the number of looped steps as 20. In [4], the authors also use a similar approach with a fixed number of loops. Such methods could be used when we do not know T(n) beforehand. However, doing so does not utilize the side information of pre-defined number of steps, while our experiments show that using such extra side information helps the model avoid learning shortcut solutions that does not length generalize. We also believe that this is a relatively weaker form of side information, compared to the other side information used in other methods such as scratchpad training.
>
> As shown in [5], there is no known way to find the number of layers even if we want the model to learn certain RASP-L programs and they say “RASP program length does not perfectly correspond to Transformer-complexity”, and we found it similar in n-RASP-L too. We treat the number of layers inside a loop as a task-dependent hyperparameter of the looped transformer models as in [5].
>
> [4] Transformers Can Do Arithmetic with the Right Embeddings
> [5] What Algorithms can Transformers Learn? A Study in Length Generalization
>
> ### Experimental confusions
> We acknowledge the typo in equation (2) and the description of the multiplication task. We will fix them in the revised draft.
> For binary addition, we tested both n and n+1 and they have similar performances, so we presented the results from n. We will also add the results from n+1 in the revised draft.
> For Section 6.1.2, we apply input injection to “NTP-Loop” since input injection is normally used as a trick for looped models. We chose the 20x larger depth to match the effective depth of the looped model, and the in-distribution performances from those NTP-based models are all near perfect, which shows that at least it is not too deep to learn (i.e., no issue with optimization). We also tried shallower models in NTP-based methods and the performance is not significantly better.
> For adding adaptive inference results: Thank you for the suggestion and we will add them in the final draft.

---

> > ### Author Response · Authors · 2024-11-21
> > **Rebuttal (2/2)**
> >
> > ### Do the trained models simulate the n-RASP-L program?
> > We read out the intermediate predictions for the parity task and found they share similar patterns with our n-RASP-L solution such that the parity location has some periodic changes. For example, for input 1010101010 (length 10, five 0’s and five 1’s alternating each other), the prediction parity read out after each step shifts 5 times between 0 and 1 and end with 1; for input with two 1’s, the output would change two times and end with 0, etc. If we visualize the intermediate embedding space, it also circles as the inference step increases. Although the intermediate reading out does not necessarily make sense, the visualization still shows that it somehow learns an iterative algorithm that is similar to the n-RASL-L program. We will add the visualization results in the appendix of the final draft as another finding.
> >
> > ### Questions on Figure 3; Other minor issues
> > About the visualization in Figure 3: Yes, we mentioned that “the inputs are preprocessed” in the caption of Figure 3.
> > About minor issues: We thank you for pointing them out and will fix them accordingly in the revised draft.

---

> > > ### Comment · Reviewer_gBs4 · 2024-11-23
> > > **Additional Response**
> > >
> > > Thank you for writing the rebuttal. Before I leave comments and questions on it, please recall that, in ICLR, the authors can revise and upload their manuscript during the discussion period. I’m not sure that the authors did so. If the authors don’t make any revisions until the end of the discussion period, I might lower my score below the acceptance because it makes me believe that they don’t have much intention to further improve their paper. The reviewers could give additional feedback from the revised version, couldn’t they?
> > >
> > > Now, let me provide my further answer. I leave no answers for the rebutted points for which I am almost satisfied.
> > >
> > > **Definition of RASP-L**
> > >
> > > - I am looking for the exact revised form of Definition 3.1. For now, it is not clear how the description of the definition will change.
> > >
> > > **Prior knowledge of the task structure**
> > >
> > > - I wonder whether the authors will also present the experimental results for FAP + fixed number of loops (like 20), as a demonstration of “ignoring the side information.”
> > > - The authors claim that the “predefined number of steps” $T(n)$ is a weaker form of side information. I’m a bit suspicious about this statement because, in order to precisely characterize a $T(n)$, it seems that the exact implementation of the $n$-RASP-L program is necessary, which requires the actual algorithm of solving the given task. In fact, this is the same for scratchpadding, although the author’s work does not precisely teach a Transformer the task-solving rule unlike scratchpad. Although the amount of information given to a model is different, I think the same amount of information is necessary to prepare the training. Considering these, can $T(n)$ still be a weaker form of side information? I would be able to admit this claim if there is a magical way to infer a proper choice of $T(n)$ without knowing the exact task-solving algorithm in the form of $n$-RASP-L program.
> > > - Extending from the point above, I express my concern because there might be many general tasks that can’t be solved with only a single loop. For example, observe that a usual algorithm (performed by humans) for solving the general *integer multiplication* doesn’t work by simply repeating the same unique job. It looks very hard to specify the “predefined number of steps” for such tasks. Thus, I want to have an author’s discussion on the cases where the predefined number of steps is not clearly known or can never be obtained; if it is a limitation of this work, the authors should make it clear in their last section of the main text.
> > >     - After pondering a bit about the multiplication (where both operands’ lengths can vary), I came up with a possible workaround: simply stacking multiple looped transformer layers, as done by McLeish et al. (2024)! If we break down the multiplication, we first do $N$-digit by $1$-digit multiplication several times (1st loop), shift their digits to the left properly (2nd loop), and add them all (3rd loop): that is, the usual algorithm may be solvable by multiple loops!
> > >     - With that in mind, if time permits, can you provide the experimental results for multi-looped transformers, especially on general (binary) multiplication, where the lengths of both numbers are the subject of length generalization? I believe that the “Binary Multiplication” task in the current paper only considers the length of the second operand. Although McLeish et al. (2024) were not really successful in achieving a significant length generalization for general integer multiplication in the NTP setup, it would be extremely interesting if the multi-looped transformer works for this task in the FAP setup.
> > >
> > > **Experimental confusions**
> > >
> > > - Although I understand that it is impossible to run all experiments in an adaptive inference setup, I’ll be happy if I can see some initial results (at least for one task). Does it show a similar performance as in the Oracle setup?
> > >
> > > ---
> > >
> > > **References**
> > >
> > > McLeish, Sean et al. “Transformers Can Do Arithmetic with the Right Embeddings.” NeurIPS 2024.

---

> ### Author Response · Authors · 2024-11-24
> **Additional Rebuttal**
>
> Thank you for the careful review and the additional response! Sorry for the delay in updating the draft. We uploaded a new version of the revised draft with changes marked in blue.
>
> **Definition of n-RASP-L**
>
> We updated the definition of n-RASP-L in Section 3, including the pre/post-processing steps as we discussed before.
>
> **Prior knowledge of the task structure & experimental confusions**
>
> **1. “I wonder whether the authors will also present the experimental results for FAP + fixed number of loops”**
>
> Thank you for the suggestions! We are currently running experiments for FAP + a fixed number of loops. Due to time constraints, it might not be available before the end of the discussion period, so we are running with a smaller scale for now. We believe that we can give an additional update within the next few days before the discussion period ends. (And later, we will be able to run the full-blown experiments.)
>
> **2. “The exact implementation of the n-RASP-L program seems necessary. Can $T(n)$ still be a weaker form of side information?” (paraphrased)**
>
> This is a great question! We believe that it’s still (slightly) weaker side information. This is because it might be possible to know the number of steps without knowing the exact algorithm. For instance, this can happen when the upper/lower bounds of computational complexity are known. Because of this, one can also simply try multiple $T(n)$ candidates (say log, linear, quadratic, …) and choose the one that performs the best on (out-of-distribution test length) validation.
>
> **3. “Although I understand that it is impossible to run all experiments in an adaptive inference setup, I’ll be happy if I can see some initial results (at least for one task). Does it show a similar performance as in the Oracle setup?”**
>
> As you suggested, we ran experiments with our adaptive stopping criterion applied and presented the result in Figure 9, Appendix D as suggested. (We are happy to move this to the main body if the reviewers think that’s a better idea – for now, we couldn’t do so due to the page limit.)
>
> **4. “Extension of n-RASP-L to support multiple loops”**
>
> Thank you for sharing the great idea with us. We did not have time to run additional experiments for this, but we agree with your idea that allowing for multiple loops in our framework can handle a much larger class of tasks including a more general length generalizable multiplication. We revised the last section (limitation & conclusion) with an additional future work direction based on your suggestion.

---

> > ### Comment · Reviewer_gBs4 · 2024-11-25
> > **Retaining my score**
> >
> > Thank you so much for revising the manuscript and providing a further response. I am mostly happy with the response and the revised paper.
> >
> > * One minor concern is a partial failure in the "FAP-Loop-Adaptive-Instance" setup, but I don't think this is a problem because individual instances may not capture the whole structure of the problem.
> > * Another minor point is that it might be better to use hats differently in Eq. (2) and equations nearby there: I recommend using $\hat{y}^t_l$ instead of $\hat{y^t_l}$.
> >
> > Given all the promises of further updates, I keep my score to 6.

---

> > > ### Author Response · Authors · 2024-11-26
> > >
> > > Thank you very much for your great suggestions.
> > >
> > > Per your suggestion “I wonder whether the authors will also present the experimental results for FAP + fixed number of steps (like 20), as a demonstration of “ignoring the side information”, we update the results from 3 tasks using full answer prediction with a fixed number of steps (20). We follow the evaluation setup in the draft and present the test accuracy of both in-distribution length and the largest length tested in Figure 4 for each task. We observe that similar to NTP+a fixed number of loops, the performance decreases without using the side information of when to stop. For reference, we also provide the test accuracy of full answer prediction with the pre-defined number of steps as $T(n)$ in the table.
> > >
> > > | Tasks                | # of steps      | In-distribution Acc | OOD Acc  |
> > > |---------------------------|-------------|-----------------|-----------|
> > > | Parity  | T(n) | 1.0±0.0         | 1.0±0.0   |
> > > |                           | Fixed (20)  | 1.0±0.0         | 0.49±0.05 |
> > > | Copy     | T(n)| 1.0±0.0         | 0.95±0.02 |
> > > |                           | Fixed (20)  | 1.0±0.0         | 0.49±0.05 |
> > > | Addition | T(n) | 1.0±0.0         | 0.99±0.01 |
> > > |                           | Fixed (20)  | 1.0±0.0         | 0.49±0.05 |
> > >
> > > We will run more extensive experiments (not just one test length, but the full evaluation as in Figure 4, and not just three tasks, but all six tasks) and incorporate them in our camera-ready version.
> > >
> > > Besides, we also updated the use of hat in Eq. (2) as suggested in the draft.
> > >
> > > Please let us know if you have any further questions!

---

### Meta-Review · Area_Chair_FF3S · 2024-12-20

**Metareview:**

This paper studies length generalization in Transformers through the lens of looped architectures, showing that Transformers with repeated layers can effectively generalize to longer sequences when trained appropriately. The reviewers appreciated the paper's clear writing, empirical evaluation, and theoretical framing of n-RASP-L.
While some reviewers raised concerns about the requirement of knowing the number of steps T(n) during training, the authors adequately addressed this through additional experiments and discussion.
I recommend acceptance, as the reviewers found the paper insightful. For the camera-ready version, I strongly suggest incorporating feedback from the discussion (e.g. clarifying the role of T(n), adding the adaptive-stopping-time results, etc).

**Additional Comments On Reviewer Discussion:**

See above.

---

### Decision · Program_Chairs · 2025-01-22

Accept (Poster)